# Genome-Wide Identification and Expression Analysis of ANS Family in Strawberry Fruits at Different Coloring Stages

**DOI:** 10.3390/ijms241612554

**Published:** 2023-08-08

**Authors:** Yongqing Feng, Shangwen Yang, Wenfang Li, Juan Mao, Baihong Chen, Zonghuan Ma

**Affiliations:** College of Horticulture, Gansu Agricultural University, Lanzhou 730070, China

**Keywords:** strawberry, *ANS* gene family, bioinformatics analysis, relative expression levels

## Abstract

To elucidate the structural characteristics, phylogeny and biological function of anthocyanin synthase (ANS) and its role in anthocyanin synthesis, members of the strawberry *ANS* gene family were obtained by whole genome retrieval, and their bioinformatic analysis and expression analysis at different developmental stages of fruit were performed. The results showed that the strawberry *ANS* family consisted of 141 members distributed on 7 chromosomes and could be divided into 4 subfamilies. Secondary structure prediction showed that the members of this family were mainly composed of random curls and α-helices, and were mainly located in chloroplasts, cytoplasm, nuclei and cytoskeletons. The promoter region of the *FvANS* gene family contains light-responsive elements, abiotic stress responsive elements and hormone responsive elements, etc. Intraspecific collinearity analysis revealed 10 pairs of *FvANS* genes, and interspecific collinearity analysis revealed more relationships between strawberries and apples, grapes and Arabidopsis, but fewer between strawberries and rice. Chip data analysis showed that *FvANS15*, *FvANS41*, *FvANS47*, *FvANS48*, *FvANS49*, *FvANS67*, *FvANS114* and *FvANS132* were higher in seed coat tissues and endosperm. *FvANS16*, *FvANS85*, *FvANS90* and *FvANS102* were higher in internal and fleshy tissues. Quantitative real-time PCR (qRT-PCR) showed that the *ANS* gene was expressed throughout the fruit coloring process. The expression levels of most genes were highest in the 50% coloring stage (S3), such as *FvANS16*, *FvANS19*, *FvANS31*, *FvANS43*, *FvANS73*, *FvANS78* and *FvANS91*. The expression levels of *FvANS52* were the highest in the green fruit stage (S1), and *FvANS39* and *FvANS109* were the highest in the 20% coloring stage (S2). These results indicate that different members of the *FvANS* gene family play a role in different pigmentation stages, with most genes playing a role in the expression level of the rapid accumulation of fruit coloring. This study lays a foundation for further study on the function of *ANS* gene family.

## 1. Introduction

Flavonoids are the most abundant polyphenolic compounds in plants and are a larger class of secondary metabolites widely distributed in the roots, stems, leaves, flowers, fruits, seeds and other organs of plants. Flavonoids have a variety of biological activities, and studies have found that they have an anticancer function [1,2]. Oxidative stress caused by reactive oxygen species produced by ROS participates in the development of inflammatory process, leading to the occurrence of many cancers, while flavonoids can regulate the activity of ROS scavenging enzymes and induce cell apoptosis, thus inhibiting the proliferation of cancer cells [3]. Other studies have found that flavonoids also have anti-inflammatory, antioxidant, antiviral and vasodilatation-inducing effects [4,5,6,7].

Anthocyanidin, a secondary metabolite of flavonoids, is a water-soluble natural pigment that provides rich colors, such as red, purple, blue and blue violet to different plant tissues and organs, attracts insects for pollination and seed dispersal as well as protects plants from various biotic and abiotic stresses [8,9]. Anthocyanins are mainly divided into the six major categories: pelargonidin, cornflower, peonidin, delphinidin, malvidin and petunia pigments [10]. Anthocyanin can reduce the liver damage caused by alcohol [11] and has anti-cancer and anti-diabetes functions [12,13]. Studies have shown that regular intake of vegetables and fruits can prevent chronic diseases [14,15,16]. Some foods rich in anthocyanins, such as small cherries and bilberries, can be used to treat atherosclerosis and improve vision, and anthocyanins are also used as colorants in the food industry [17]. Genome-wide identification is helpful to identify color-specific genes and provide a theoretical basis for improving strawberry quality [18].

Anthocyanin synthase (ANS) is an enzyme located in the downstream of anthocyanin synthesis pathway, plays an important role in anthocyanin synthesis pathway, catalyzes the conversion of colorless anthocyanin to anthocyanin and belongs to the 2-oxyglutarate iron-dependent oxygenase superfamily [19]. Basil (*Ocimum basilicum*) contains two homologous *ANS* genes, but each gene carries a loss-of-function mutation that gives the basil its green color, but some purple basil is due to *ANS1* being functional, while *ANS2* carries a nonsense mutation that introduces the *ANS* gene into Arabidopsi. T-DNA was inserted into the *ANS* gene and it was found that anthocyanins did not accumulate in Arabidopsis mutants [20]. Previous studies on Zoysiagrass (*Zoysia japonica* Steud) found that the expression of *ZjANS* gene was upregulated in purple spike tips and stolons, while the expression of *ZjANS* gene was low in green varieties [21]. In eggplant, it was found that the expression of *ANS* gene increased gradually with the increase of development stage, and the expression level was high in purple genotype, which was positively correlated with the content of anthocyanin [22]. The defects or silences of *ANS* gene function can affect the formation of plant and fruit color. The low content of anthocyanin in yellow raspberry (*Rubus idaeus*) is due to a 5 bp insertion in the *ANS* gene coding region, which produces a premature stop codon. This mutation leads to the loss of *ANS* gene function. The anthocyanin content of yellow raspberry decreased significantly [23]. The *BjANS* gene of Brassica juncea was expressed in both black seed coat and embryo, while in yellow seed coat, the deletion of *BjANS* gene expression prevented the biosynthesis of proanthocyanidins, so the seeds were yellow due to the transparency of the seed coat [24]. Eun-Young Kim et al. [25] identified two new allelic variants of *ANS* genes, which could cause the deletion of onion anthocyanin synthesis.

*ANS* is an important regulatory gene for anthocyanin synthesis and plays an important role in fruit coloring [26]. At present, there are many reports about the *ANS* gene family, but the number of members of this family in strawberries and the expression of each member in the fruit coloring process have not been reported. Based on this, this study identified members of the strawberry *ANS* gene family from the whole genome of strawberries, and conducted bioinformatics analysis. Real-time quantitative PCR (qRT-PCR) was used to analyze the changes of expression levels of each member in four different coloring stages of strawberries (S1: green fruit stage; S2: 20% coloring stage; S3: 50% coloring stage; S4: complete coloring stage) to elucidate the role of each member in regulating anthocyanin synthesis in fruit and provide candidate genes for further research on its function in the future.

## 2. Results

### 2.1. Identification and Physicochemical Property Analysis of the ANS Gene Family in Strawberries

Using the amino acid sequence of the Arabidopsis *ANS* gene as the query sequence, a total of 141 genes were retrieved using the TBtools blast alignment and NCBI protein blast, named *FvANS1-FvANS141* based on the position of the gene on the chromosome. The shortest amino acid length was 151aa (*FvANS29*) and the longest was 608aa (*FvANS20*). The molecular weight ranged from 17,122.77 Da to 68,294.08 Da. The isoelectric point ranged from 4.77 (*FvANS92*) to 9.65 (*FvANS29*). *FvANS29*, *FvANS52*, *FvANS73*, *FvANS85*, *FvANS93*, *FvANS94* and *FvANS110* are basic proteins with isoelectric points greater than 7, and the rest are acidic proteins (Appendix A).

### 2.2. Evolutionary Tree, Secondary Structure and Subcellular Localization of the ANS Family in Strawberries

The amino acid sequences of 141 *ANS* genes of strawberries were used for phylogenetic analysis (Figure 1). According to the evolutionary relationship, there were four subfamilies, with a maximum of 46 members in group4 and a minimum of 24 members in group3. Secondary structure prediction (Appendix A) showed that all genes had no β-corners, mainly α-helix, random curling and extended chains, among which most were random curling (13.82–59.58%), followed by α-helix (18.07–50.68%) and the lowest percentage were extended chain (9.94–26.51%). Subcellular localization prediction results showed (Figure 2) that the *ANS* gene family members were mainly located in chloroplasts, cytoplasm, nuclei and cytoskeletons. A total of 32 genes were located in mitochondria, 20 genes were located in vacuoles, 23 genes were located in Golgi apparatus and only 9 genes were located in endoplasmic reticulum.

### 2.3. Gene Structure, Motif, Domain and Promoter Cis-Acting Elements

According to gene structure analysis (Figure 3), the 36 *FvANS* genes did not contain upstream and downstream sequences, and the number of exons was 1–11. *FvANS68, FvANS108, FvANS117* and *FvANS130* contained only 1 exon, while *FvANS12, FvANS17* and *FvANS96* contained 11 exons. Most genes contain 2–4 exons and have the same distribution and length in the gene structure of the same subfamily gene. The conserved motif of *FvANS* gene family proteins is predicted at the MEME website (Figure 3), containing a total of 15 motifs; most of the N terminus is motif12, the C terminus is motif8. *FvANS20* contains 2 motif5, *FvANS40* contains 2 motif6, *FvANS71*, *FvANS73*, *FvANS76* and *FvANS78* contain 2 motif13, and motif11 exists only in group1. The *FvANS* genes were all found to have the 2OG-FeII_Oxy domain at NCBI-CDD prediction (Figure 3). Cis-acting elements play an important role in transcriptional regulation of genes. A total of 43 major cis elements were predicted in the *FvNAS* gene promoter region (Figure 4). Analysis showed that *FvANS* gene mainly contained light, hormone, abiotic stress and tissue-specific and development-related elements. Among them, the light-response elements include GT1-motif, G-box, Box II and I-box. Hormone-response element contains AuxRR-core (auxin-response element), GARE (gibberellin-response element), ABRE (abscisic-acid-response element), TCA element (salicylic-acid-response element) and ERE (ethylene-response element). Stress response elements include LTR (low-temperature-response element), W-box, MYB and ARE (anaerobic cis-regulatory element). The primary tissue-specific and development-related cis elements included RY repeat (seed-specific regulatory response element) and CAT-box (meristem-expression-response element).

### 2.4. Chromosomal Localization and Collinearity Analysis

TBtools was used for chromosome localization analysis, and 141 members were distributed on 7 chromosomes. There are 18 genes on chromosomes 1 and 3, 34 genes on chromosomes 2, 19 genes on chromosomes 4, 5 and 6, and 14 genes on chromosome 7. The genes on chromosome 2 distributed the most, accounting for 24.11% of the total genes, and the genes on chromosome 7 distributed the least, accounting for 9.93% of the total genes (Figure 5). To further understand the evolutionary relationships of gene families, intra- and inter-species collinearity analyses were performed with the MCScanX tool of TBtools. A total of 10 collinear relationships were found in *FvANS* gene family species (Figure 6), which were located on chromosomes chr1, chr2, chr4, chr5, chr6 and chr7, namely *FvANS18/FvANS86*, *FvANS31/FvANS80*, *FvANS35/FvANS74*, *FvANS37/FvANS72*, *FvANS44/FvANS121*, *FvANS48/FvANS124*, *FvANS91/FvANS135*, *FvANS109/FvANS136*, *FvANS131/FvANS138* and *FvANS135/FvANS141*, respectively. *FvANS135* has two tandem repeats, and these results suggest that some *FvANS* genes may arise through gene duplication, which may have similar functions.

The collinear relationships between strawberries and *Arabidopsis thaliana*, grapes, apples and rice were plotted respectively (Figure 6). The homologous genes with *Arabidopsis thaliana*, grape, apple and rice were 42, 59, 102 and 18 pairs, respectively, indicating that strawberries and dicotyledon plants had more homologous genes than monocotyledon plants.

### 2.5. Codon Preference and Selection Pressure Analysis

The components of the codons include CAI (codon adaptation index), CBI (codon bias index), Fop (frequency of optical codons), Nc (effective number of codon), GC (guanine and cytosine), GC1 (GC at the first codon position), GC2 (GC at the second codon position) and GC3 (GC at the third codon position), etc. The use frequency of relative synonymous codons in the strawberry genome was analyzed. It was found that RSCU of the following 33 codons was ≥1: UUC, UUG, UCU, UCA, UAC, UAA, UGC, UGA, CUU, CUC, CCU, CCA, CAU, CAA, AUU, AUC, AUG, ACU, ACC, ACA, AAU, AAG, AGC, AGA, AGG, GUU, GUG, GCU, GCA, GAU, GAG, GGU and GGA. Among them, 11 of the third codons are U, 9 are A, 7 are C and 6 are G, indicating that the third codon of the amino acid of the strawberry ANS protein is more inclined to U or A. Among the 33 codons, the third codon is 10593(U), 5885(A), 5442(C) and 7434(G), accounting for 36.09%, 20.05%, 18.54% and 25.33% of the total codon, respectively (Figure 7). The mean values of CAI, CBI, Fop and Nc in strawberry *ANS* family members were 0.209, −0.038, 0.393 and 55.02, respectively. The GC content of *FvANS* family members ranged from 40.80% to 59.70%, and the GC3s content ranged from 33.40% to 76.60%, with the mean values of GC and GC3s being 45.86% and 46.03%, respectively. A total of 13 genes were found to have Nc values less than 50, they are *FvANS2, FvANS3, FvANS8, FvANS15, FvANS23, FvANS42, FvANS62, FvANS85, FvANS88, FvANS97, FvANS104, FvANS112* and *FvANS132*, respectively. This indicates that the codon preference of these 13 genes is strong (Figure 8). Correlation analysis showed (Figure 9) that T3s was positively correlated with A3s, and negatively correlated with C3s, G3s, CAI, CBI, Fop, Nc, GC and GC3s. Furthermore, C3s and G3s were negatively correlated with T3s and A3s, and positively correlated with CAI, CBI, Fop, GC and GC3s. Nc was negatively correlated with T3s and A3s, but positively correlated with C3s, GC and GC3s.

The Ka/Ks allowed estimation of their evolutionary selection pressure to further understand the evolutionary relationships of the strawberry *ANS* gene family (Table 1). From 10 pairs of genes with collinear relationship, the Ka/Ks of 6 pairs of genes were calculated to be less than 1, suggesting that the strawberry *ANS* gene family may be dominated by purifying selection.

### 2.6. Expression Pattern Analysis and Protein Interaction Prediction

Analysis of the expression patterns of *FvANS* gene family members throughout plant development, including seeds (ovary wall, embryo, endosperm and seed coat tissue), young leaves, seedlings, different tissues in flowers (perianth, carpels, inner pellicle and fleshy tissue below achene), and pollen (and pollen microspore), found similar expression levels of genes in the same subfamily (Figure 10). The expression level of *FvANS9* was higher in leaves, but lower in other tissues. The expression levels of *FvANS11* and *FvANS117* were higher in the ovary wall of stage 2, stage 3, stage 4 and stage 5. *FvANS102*, *FvANS117* and *FvANS126* were highly expressed in style. *FvANS15*, *FvANS41*, *FvANS47*, *FvANS48*, *FvANS49*, *FvANS67*, *FvANS114* and *FvANS132* were highly expressed in seed coat tissue and endosperm of stage 3, 4 and 5. *FvANS16*, *FvANS85*, *FvANS90* and *FvANS102* were highly expressed in the inner tissue and fleshy tissue of the receptor. *FvANS53* was highly expressed in the carpel. *FvANS47* and *FvANS120* were highly expressed in the seeds.

The interactions among 141 FvANS proteins were predicted by the STRING online website (Figure 11). The results showed that 59 FvANS proteins may interact with each other, and the 59 FvANS proteins interact with each other to form a protein interaction network (PPI network). Most FvANS proteins form a complex network structure, such as FvANS10, FvANS16, FvANS40, FvANS41, FvANS48, FvANS62, FvANS64, FvANS90, FvANS131, FvANS132, etc. There are also some FvANS proteins that only have a single interaction, such as FvANS6 and FvANS130, FvANS96 and FvANS118, FvANS53 and FvANS134, FvANS43 and FvANS58, etc. The three-dimensional structure of the FvANS53 and FvANS90 proteins is known. FvANS16, FvANS137, FvANS139, FvANS125, FvANS70, FvANS128, FvANS141, FvANS40, FvANS90, FvANS136, FvANS17, FvANS141 and FvANS115 are associated with XP_004299308.1(Flavonoid 3′-monooxygenase-like) interaction. FvANS16, FvANS53, FvANS90, FvANS136, FvANS40 and FvANS109 interacted with XP_004294725.1 (Trans-cinnamate 4-monooxygenase-like). FvANS16, FvANS70, FvANS90 and FvANS40 interact with XP_004307864.1 (4-coumarate-CoA ligase 2-like). FvANS70 interacts with XP_004310219.1 (4-coumarate-CoA ligase-like 1-like). FvANS16, FvANS40, FvANS90 and FvANS141 interact with XP_004307734.1 (Chalcone-flavonone isomerase 3). The interaction between FvANS16, FvANS17, FvANS70, FvANS90, FvANS116, FvANS128 and FvANS136 with XP_004309662.1 (Anthocyanidin reductase-like). The interaction of FvANS16, FvANS40, FvANS90, FvANS109, FvANS136 and FvANS141 with XP_004307451.1 (Chalcone-flavonone isomerase 1-like). FvANS16, FvANS17, FvANS40, FvANS70, FvANS90, FvANS109, FvANS115, FvANS128, FvANS136 and FvANS141 and XP_004291858.1 (Bifunctional dihydroflavonol 4-reductase) interaction. The interaction between FvANS90 and XP_004297144.1 (Leucoanthocyanidin reductase-like).

### 2.7. Determination of Anthocyanin Content in Strawberries at Different Coloring Stages and Expression Analysis of FvANS Gene Family

S1 to S4 are different coloring stages of strawberries, which are green fruit stage, 20% coloring stage, 50% coloring stage and complete coloring stage, respectively (Figure 12). With the increase of fruit coloring, anthocyanin content increases gradually. qRT-PCR analysis (Figure 13) showed that *FvANS* gene was expressed at all stages, indicating that the *ANS* gene family may be involved in each stage of strawberry color transformation, but the expression level showed irregular changes at different growth stages. The expression levels of most genes were highest in S3, indicating that *FvANS* gene was highly expressed during the rapid accumulation of fruit pigment. The expression level of *FvANS114* in S3 was 2891 times that in S1. The expression of *FvANS19* in S3 was 200 times that in S1. The expression level of *FvANS78* in S3 was 174 times that in S1. The expression level of *FvANS89* in S3 was 134 times that in S1. The expression level of *FvANS16* in S3 was 83 times that in S1. The expression level of *FvANS91* in S3 was 56 times that in S1. The expressions of *FvANS10*, *FvANS39*, *FvANS76*, *FvANS109* and *FvANS122* were the highest in S2 period. Compared with S1, the S2 phase of *FvANS39* is significantly 10.77 times that of S1 phase. The expression level of *FvANS109* in S2 was 3.7 times that in S1. The expression level of *FvANS52* was highest in the S1 period, indicating that this gene plays a role in promoting anthocyanin accumulation in the S2, S3 and S4 periods.

## 3. Discussion

The *ANS* gene plays an important role in plant growth and development and adaptation to various environmental conditions. *ANS* gene has been studied and cloned in many crops, such as Arabidopsis, apple, tobacco, ginkgo biloba, etc. [27,28,29,30]. The *ANS* gene is a structural gene that regulates the phenylalanine metabolism pathway and plays an important role in plant and fruit coloration [31]. In this study, a comprehensive analysis of genome-wide identification, phylogenetic relationships, gene structure, conserved motifs, chromosomal location, co-lineage relationships, evolutionary selection pressure, codon usage bias, cis-acting elements and expression patterns of *ANS* genes in strawberries was performed. A total of 141 *ANS* genes were identified in the strawberry genome, which is a large gene family. There were more members of this family in strawberries than in tobacco (2), Arabidopsis (1) and Ginkgo (1). Physicochemical property analysis showed that most *FvANS* genes were acidic proteins, and only seven genes were basic proteins (Appendix A). Based on systematic cluster analysis, 141 strawberry *ANS* genes were classified into four subclades. Each *FvANS* gene contained conserved domain 2OG-FeII_Oxy (Figure 3). Chen Lijing et al. [32] found in Oriental Lily that *LiANS* gene belongs to the 2OG-FeII_Oxy supergene family, which is consistent with the results of this study. Subcellular predictions found that members of the *FvANS* family were mainly localized in the cytoplasm, chloroplasts and nucleus (Figure 2), which is consistent with the *ANS* reported in Arabidopsis [27], grape [33,34] and other species, as well as the conclusion that anthocyanins are synthesized mainly in the cytoplasm. Therefore, we speculate that *ANS* genes are involved in the synthesis of strawberry anthocyanins.

Gene duplication plays an important role in the evolution of organisms, including tandem duplication, local duplication and whole genome duplication [35]. Chromosomal localization revealed that 141 genes were distributed on 7 chromosomes (Figure 5). Some genes formed gene clusters on chromosomes, and it is speculated that this may be formed by tandem repeats, suggesting that tandem repeats may be a major cause of *ANS* family amplification in strawberries. There were 10 collinear relationships in the *FvANS* gene family, namely, *FvANS18/FvANS86*, *FvANS31/FvANS80*, *FvANS35/FvANS74*, *FvANS37/FvANS72*, *FvANS44/FvANS121*, *FvANS48/FvANS124*, *FvANS91/FvANS135, FvANS109/FvANS136*, *FvANS131/FvANS138* and *FvANS135/FvANS141*, respectively. *FvANS135* has two tandem repeats, all of these gene pairs were in the same family except for *FvANS31*/*FvANS80* and *FvANS44*/*FvANS121.* In the study of forest strawberries and pineapple strawberries, Guo Lili et al. [36] found that each pair of genes belonged to the same subfamily with high homology, and their gene structure and conserved motifs were very similar, suggesting that these genes may have similar functions (Figure 6). Gene selection pressure and codon preference analysis are also useful for understanding their evolutionary relationships. In this experiment, performed selection pressure analysis was performed and found that the strawberry *ANS* gene was mainly selected for purification (Table 1). In addition, codon preference analysis of the strawberry *ANS* family showed weak codon preference for the strawberry *ANS* genes (Figure 7 and Figure 8). The promoter of a gene may determine the function of a gene. Studies have shown that there is a light signal recognition element G-box in the promoter region of the structural gene of anthocyanin synthesis, which is regulated by light and participates in anthocyanin biosynthesis [37]. ABA-response element (ABRE) was shown to be involved in the anthocyanin biosynthesis process in lychee pericarp [38]. In this study, cis-acting elements were analyzed in the 2000 bp sequence before the transcription start site of *FvANS* gene, and it was found that there were more response elements in response to light, hormone and abiotic stress. Most of the *ANS* genes contain GT1-motif, G-box, ABRE, GARE, ARE and other functional elements (Figure 4), indicating that the *ANS* gene family is significantly related to the regulation of anthocyanin synthesis.

Flavonoids, like other secondary metabolites, accumulate mostly with tissue properties, and key genes involved in the synthesis of these metabolites also have tissue expression specificity, which has been widely used in the isolation and identification of key genes of secondary metabolite synthesis pathways in genomically complex or non-model plants [39,40]. *FcANS1* transcripts were only expressed in root tips, terminal buds, young leaves and young stems of fig trees, but not in mature leaves, stems or petioles [41]. Yue et al. [42] found that the expression of *CmANS* gene in Chuzhou Chrysanthemum was the highest at the blooming stage and the lowest at the budding stage. In Clivia, *CmiANS* were found to be expressed at a higher level in perianth than in other tissues (leaves, styles, stigmas and flower stems) [43]. In sweet potato, *IbANS* was expressed in all tissues, but the expression level was higher in root tuber and pericarp [44]. In potato, the expression level of *StANS* gene in color genotype was significantly higher than that in yellow genotype, and its content was the highest in purple epiderm. Overexpression of *StANS* gene could promote anthocyanin synthesis in potato tuber [45]. The expression level of *SmANS* gene was the highest in eggplant peel and the lowest in root and pulp, and the expression of *SmANS* gene was completely dependent on light [46]. Based on molecular analysis, this study analyzed the expression levels of *FvANS* gene in different tissues. The members of the *ANS* gene family were expressed in different tissues, and their members played various important functions. Most of the genes were expressed at low levels in various tissues, but some of them were tissue specific (Figure 10). For example, the expression of *FvANS9* was higher in leaves and lower in other tissues. The expressions of *FvANS11* and *FvANS117* were higher in the ovary wall of stage 2, stage 3, stage 4 and stage 5. *FvANS102*, *FvANS117* and *FvANS126* were highly expressed in style. *FvANS15*, *FvANS16*, *FvANS41*, *FvANS47*, *FvANS48*, *FvANS49*, *FvANS67*, *FvANS114* and *FvANS132* were highly expressed in ghost (seed coat and endosperm) of stage 3, 4 and 5. The expression levels of *FvANS16*, *FvANS85*, *FvANS90*, *FvANS102* and *FvANS126* were highly expressed in the strawberry cortex and pith tissues. In peaches, it was found that *PpANS* gene expression was highest in peel, highest in pulp, weaker in flowers, lowest in leaves and not measured in roots and stems, and that *PpANS* gene was positively correlated with anthocyanin content in fruits [47]. *FvANS53* was highly expressed in pericardium, perianth, receptacle and anther. Pericarp, pulp and seed coat are all highly pigmented parts. Consistent with the above results, the gene was highly expressed in the parts with high anthocyanin content.

Previous studies found that proteins are grouped according to their cellular rather than molecular function, so unidentified protein interactions can reasonably predict the cellular function of proteins [48]. Protein interaction prediction results showed that some genes were associated with Flavonoid 3′-monooxygenase, 4-coumarate CoA ligase, Trans-cinnamate 4-monooxygenase, Bifunctional dihydroflavonol 4-reductase, Chalcone—flavonoid isomerase 3, Chalcone synthetase 1 and Leucoanthocyanidin reductase, which may jointly regulate the synthesis of anthocyanins (Figure 11). Kumari et al. [49] found that flavonoid 3′-monooxygenase was mainly involved in the degradation of kaempferol. Studies have found that 4-coumaric acid CoA ligase is closely related to apple skin coloring [50]. Watanabe et al. [51] used CRISPR/Cas9 technology to target mutate the DFR-B gene of Japanese petumnia and achieved the transformation of stem color from purple to green and petals from purple to light pink in the transgenic plants. In *Ginkgo biloba*, it was found that the activity of chalcone isomerase and the expression of chalcone isomerase gene were related to the accumulation of flavonoids in leaves [52]. In *Oncidium hybridum*, the inactivation of chalcone synthase resulted in the inability of floral organs to accumulate anthocyanin [53].

It was found that the synthesis of *ANS* was closely related to the content of anthocyanins [54]. The main cause of anthocyanin deficiency in the petals of *Forsythia admirata* may be the non-expression or low expression of *ANS* gene [55]. NAKAMURA et al. [56] inhibited the expression of *ANS* gene in the petals of the herb Torenia fournieri by RNAi technology, and the flower color changed from blue to white, and the white flower character could be stably inherited. Aharoni et al. [57] inhibited the expression of *ANS* gene in strawberries, which significantly reduced anthocyanin accumulation and turned the corolla from pink to white. REDDY et al. [58] increased anthocyanin content in transgenic plants through over expression of *ANS* gene in rice breeding, resulting in purplish red seed coat of rice. The total anthocyanin content of ANS L15 and ANS L18 in the three *ANS* gene transfer lines of strawberries was significantly increased compared with the control [59]. Deletion of the *SmANS* gene caused the mutation of red Miltiorrhiza color into white flowers [18]. *ANS* may be a limiting factor in the skin coloring of early white honey of Yunnan red pear (*Pyrus pyrifolia*) [60]. In mulberry, overexpression of *MaANS* significantly increased the accumulation of total flavonoids and anthocyanin in corolla [61]. In jade orchids, real-time fluorescence quantitative analysis showed that *MsANS* transcripts were present in all color petals, but the expression level was highest in red petals, where the transcription level of *MsANS* was 26 times higher than that in white petals [62]. In this study, qRT-PCR was used to analyze the expression of 48 *FvANS* in strawberries at 4 different coloring stages (Figure 13). qRT-PCR analysis showed that the expression of *FvANS52* was highest in S1 period, during which the fruit did not begin to discolor, suggesting that this gene may play an important role in the initiation of fruit staining. Similar to the previous study in mango, the expression of *ANS* gene was the highest in green peel, followed by red peel, and the lowest in yellow peel [63]. During S2-S3 period, fruits began to be colored and anthocyanin began to accumulate rapidly. Quantitative results showed that the expression levels of *FvANS10*, *FvANS39*, *FvANS76*, *FvANS109* and *FvANS122* were the highest in S2. The expression levels of *FvANS16*, *FvANS19*, *FvANS78*, *FvANS89*, *FvANS91* and *FvANS114* were the highest in S3. The genes with high expression levels in S2 and S3 indicated that these genes played an important role in the fruit coloring process. In the study of peony leaf color, *ANS* gene showed a downward trend in the purple to green stage of the leaves [64]. In different color varieties of tomato (*Solanum lycopersicum*), the total anthocyanin content of ripe purple tomato fruit was higher, and the expression level of *ANS* was also higher [65]. Consistent with the results of this study, the expression of *ANS* gene was gradually upregulated with the accumulation of pigment. The expression of most *FvNAS* genes was significantly downregulated during S4 period, such as *FvANS5*, *FvANS16*, *FvANS19*, *FvANS38*, *FvANS78* and *FvANS114*, indicating that anthocyanin synthesis in fruit peel gradually decreased during fruit ripening. Zhang Xiaodong et al. [60] find in the study of red pears that the expression of *ANS* gene in ‘Zaobaimi’ variety presents a downward trend with the maturity of fruit, which is similar to the result of this study. The strawberry *ANS* gene family is a large family, and some genes function only during specific periods, such as the *FvANS52* with the highest expression during the S1 period. At present, the detailed function of the *FvANS* gene has not been verified, and these pigment-related functions should be further verified in plant systems.

## 4. Materials and Methods

### 4.1. Plant Materials and Treatments

Strawberry fruits were used as research materials, and fruits at green fruit stage, 20% coloring stage, 50% coloring stage and fully coloring stage were collected (Figure 14), accurately weighed, quickly frozen with liquid nitrogen and stored at −80 °C for subsequent experiments.

### 4.2. Extraction and Quality Control of RNA from Strawberry Fruit

Strawberry RNA was extracted using the CTAB method. RNA quality and quantity were determined using a Pultton P200 Micro Volume Spectrophotometer (Pultton Technology Ltd., San Jose, CA, USA). RNA was stored at −80 °C for further analysis.

### 4.3. Identification of ANS Gene Family in Strawberries

The protein sequence of *ANS* genes were obtained from *Arabidopsis thaliana* database (https://www.arabidopsis.org/, accessed on 14 March 2023). Strawberry genome and annotation information were downloaded from the phytozome v13 (https://phytozome.jgi.doe.gov/pz/portal.htm, accessed on 16 March 2023) [66]. All the protein sequences of grape were extracted and aligned with the *AtANS* family protein sequence using TBtools (version 1.108) software, initial acquisition of grape *ANS* family members [67]. Then, we compared these initially screened protein sequences through the Protein Blast plate of NCBI, and analyzed the conserved domains of the protein using the NCBI-CDD website (https://www.ncbi.nlm.nih.gov/cdd/, accessed on 17 March 2023). Redundancy was removed, and the protein sequence containing the special domain 2OG-FeII_Oxy(pfam03171) of the *ANS* gene was retained. The molecular weight (MW), isoelectric point (PI), instability coefficient, fat index, and hydrophilicity of the grape *ANS* family were analyzed from the online software ExPASy (https://web.expasy.org/protparam/, accessed on 20 March 2023) [68].

### 4.4. Phylogenetic Evolution, Secondary Structure and Subcellular Localization

The multiple sequence alignment of the FvANS proteins was conducted using the ClustalX 1.83 software. In MEGA 7.0 software, the neighbor-joining method (NJ) was used to construct an evolutionary tree, and the bootstrap value was 1000, using EVOLVIEW website (https://evolgenius.info//evolview-v2/#login, accessed on 24 March 2023) for beautification [69]. The NPS@: SOPMA website (https://npsa-prabi.ibcp.fr/cgi-bin/npsa_automat.pl?page=npsa_sopma.html, accessed on 26 March 2023) was used to predict the secondary structures of FvANS proteins. The online software WoLF PSORT (https://wolfpsort.hgc.jp/, accessed on 29 March 2023) was used to predict the subcellular localization of the FvANS proteins [70].

### 4.5. Analysis of Gene Structure, Motif, Domain, and Cis-Acting Elements

Gene structure prediction was constructed using TBtools software. The conserved motifs of proteins were constructed by the MEME (http://meme-suite.org/tools/meme, accessed on 4 April 2023), the number of motifs was set to 10 and the remaining parameters were all default values [71]. The conserved domains of the protein were analyzed at the NCBI-CDD website (https://www.ncbi.nlm.nih.gov/cdd/, accessed on 10 April 2023). The sequence 2000 bp upstream of transcription start site (TSS) of each *FvANS* was extracted from the Phytozome database and analyzed using the online software New PLACE (https://www.dna.affrc.go.jp/PLACE/?action=newplace, accessed on 14 April 2023) [72] and mapped in TBtools (version 1.108).

### 4.6. The ANS Gene Location and Synteny Analysis

Chromosome localization of strawberry *ANS* family members was performed using Tbtools (Version 1.108) software. To analyze the collinearity relationships of *FvANS* genes, the genome and annotation files of arabidopsis, apple, grape and rice used for collinearity analysis were downloaded from phytozome v13 (https://phytozome.jgi.doe.gov/pz/portal.html, accessed on 18 April 2023), and the gene pairs of the *ANS* genes were determined using TBtools synteny, and the diagram was drawn via TBtools (Version 1.108) [73].

### 4.7. Codon Bias and Selective Pressure Analysis

The codon usage characteristics of the CDS sequence of *FvANS* genes were analyzed using the online software CodonW 1.4.2 (http://codonw.sourceforge.net, accessed on 20 April 2023), including relative synonymous codon usage (RSCU), effective codon (ENC), codon bias index (CBI), codon adaptation index (CAI), optimal codon usage frequency (Fop), T3s, C3s, A3s, G3s, With T3s, C3s, A3s, G3s, CAI, CBI, Nc, Fop, GC, GC3s, L_sym, L_aa, GRAVY and Aromo parameter correlation analysis. Using TBtools’ NG method, Ka (non-synonymous replacement rate), Ks (synonymous replacement rate) and Ka/Ks (selection intensity) were calculated for 10 pairs of *FvANS* genes with collinear relationship.

### 4.8. Expression Pattern and Protein Interaction Analysis of ANS Gene Family in Strawberries

The expression levels of *ANS* gene in different tissues of strawberries were retrieved in BAR database (https://bar.utoronto.ca/, accessed on 23 April 2023), including pollen, anther, style, fleshy tissue, flower, receptol, carpels and leaves, etc. The selected data were transformed by log10, and the plots were performed in TBtools. The protein interaction network was predicted by STRING Version 11 (https://string-db.org/, accessed on 25 April 2023) [74].

### 4.9. Determination of Anthocyanin Content in Strawberry Peel during Different Developmental Periods

The 1.0 g fruit was accurately weighed, ground in liquid nitrogen, put into a 10 mL centrifuge tube, rinsing the mortar with 1% HCl-methanol solution, and transferred to the test tube. The volume was fixed to the scale, and then mixed. Extraction was carried out at 4 °C for 20 min in the dark, during which the extraction was shaken several times. Samples were then filtered through 0.2 μm PES filters (Krackeler Scientific, Inc., Albany, NY, USA) and analyzed using TU-1900 double beam UV-visible spectrophotometer (Beijing Purkinje General Instrument Co. Ltd., Beijing, China). The solution was zeroed with 1% HCl-methanol solution as blank reference, and the absorbance of the solution was determined with filtrate at 600 nm and 530 nm, respectively, and repeated three times. Anthocyanin content (U) was expressed by the difference of absorbance value at wavelength 530 nm and 600 nm per gram of fresh weight peel tissue, i.e., U = (OD_530_ − OD_600_)/gFW.

### 4.10. qRT-PCR Analysis

The primers (Appendix A) were synthesized by Shanghai (Shanghai, China) Biological Engineering Co., Ltd. RNA was extracted from pinot noir fruit, reverse transcribed as single-stranded cDNA as template. The GAPDH of strawberries was used as the internal reference gene, and the quantitative reaction system was 20 μL: 1 μL cDNA, 1 μL each of upstream and downstream primers (10 μmol/L), 10 μL SYBR enzyme, 7 μL ddH_2_O. The reaction procedure was: 95 °C predenaturation for 30 s; 95 °C denaturation for 10 s, 60 °C annealing for 30 s, 72 °C extension for 30 s, 40 cycles; the test was repeated 3 times. Then the reaction procedure and the melting curve and the fluorescence value change curve were analyzed.

### 4.11. Statistical Analysis of the Data

All the data were normalized to those of actin, which served as an internal reference gene, and the relative expression of all the evaluated *FvANS* genes was calculated using the 2^−ΔΔCt^ method [75].

After normalization of the data from three independent experiments, three repeated qRT-PCR quantitative data and anthocyanin content data were analyzed using the Duncan method with one-way ANOVA in SPSS 22.0. *p* < 0.05 was significant difference and drew with Origin 2021.

## 5. Conclusions

In this study, 141 *ANS* genes of strawberries were found and distributed in 7 chromosomes, which could be divided into 4 subfamilies according to the evolutionary relationship. Promoter cis-acting element analysis revealed that the *FvANS* gene contains a number of response elements related to anthocyanin synthesis, such as light-response elements, auxin, gibberellin, abscisic acid, salicylic acid and ethylene-response elements. Protein interaction prediction results showed that some genes were associated with Flavonoid 3 ‘-monooxygenase, 4-coumarate CoA ligase, Trans-cinnamate 4-monooxygenase, Bifunctional dihydroflavonol 4-reductase, Chalcone-flavonoid isomerase 3, Chalcone synthetase 1, Leucoanthocyanidin reductase, which may jointly regulate the synthesis of anthocyanins. The results of qRT-PCR showed that the expressions of *FvANS16*, *FvANS19*, *FvANS78*, *FvANS89*, *FvANS91* and *FvANS114* were higher in the fast-coloring stage, *FvANS52* was higher in the early color turning stage. These genes can be used as candidate genes for further functional studies. Based on this study, new insights are provided for further study on the function of *ANS* gene family in promoting strawberry coloration.

## Figures and Tables

**Figure 1 ijms-24-12554-f001:**
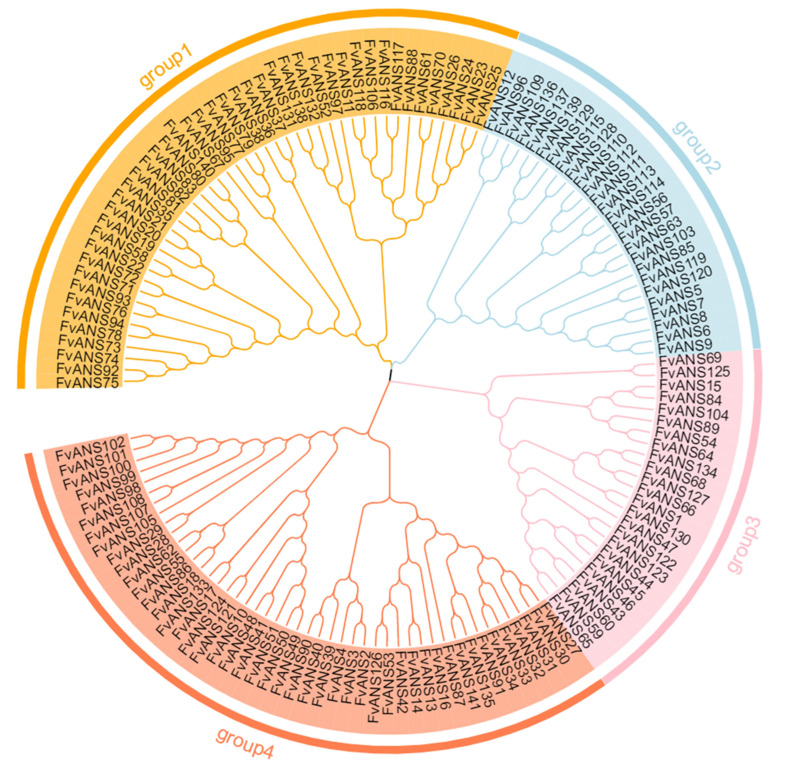
Phylogenetic analysis of the strawberry *ANS* gene family. Phylogenetic trees were constructed using the ANS protein sequences. NJ method was adopted, and the bootstrap value was set to be equal to 1000.

**Figure 2 ijms-24-12554-f002:**
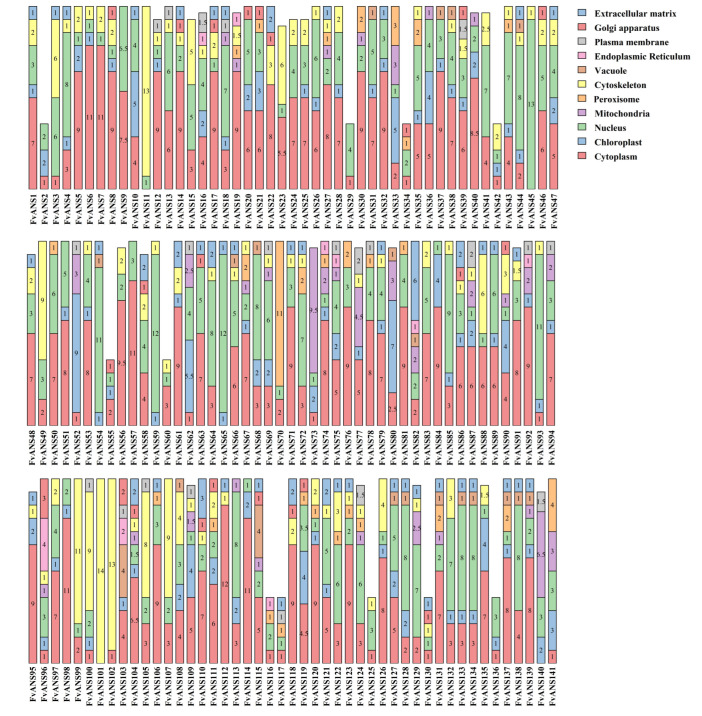
Strawberry ANS protein subcellular localization. The numbers in the bars indicate the number of subcellular localizations.

**Figure 3 ijms-24-12554-f003:**
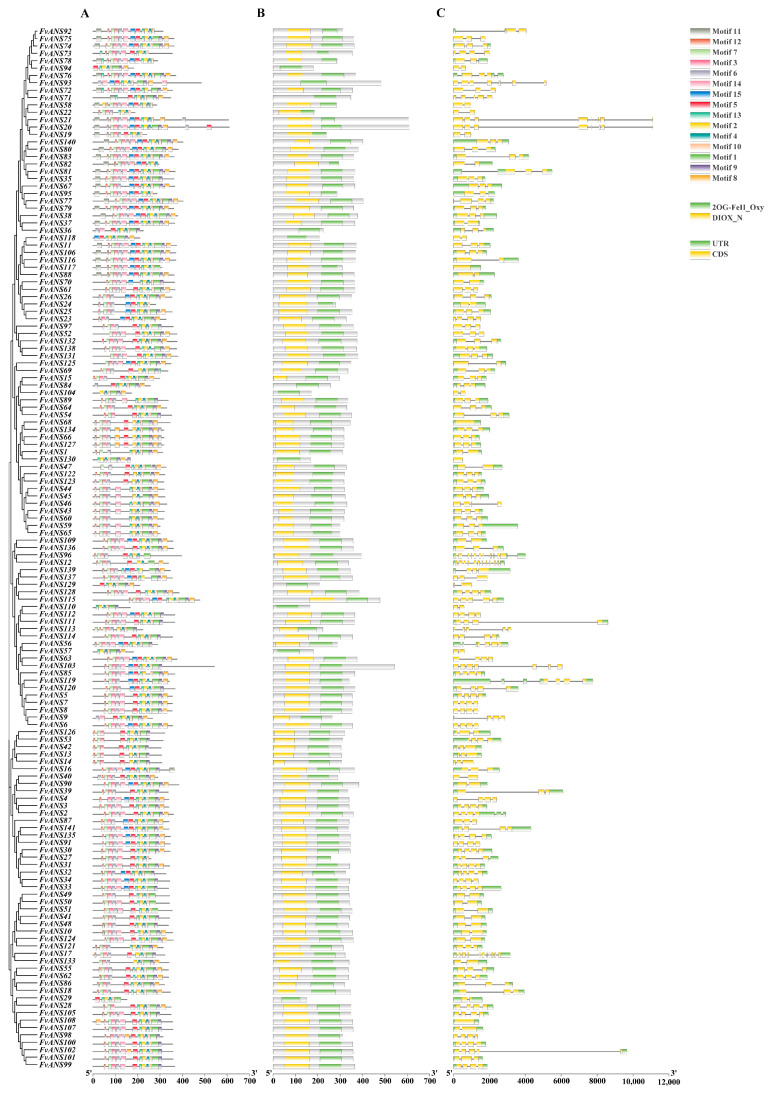
Motif, domain and gene structure analysis of *FvANS* gene. (**A**) Analysis of conserved motif of *ANS* gene in strawberries. (**B**) Analysis of conserved domain of *ANS* gene in strawberries. (**C**) The exon–intron structure of *FvANS* genes.

**Figure 4 ijms-24-12554-f004:**
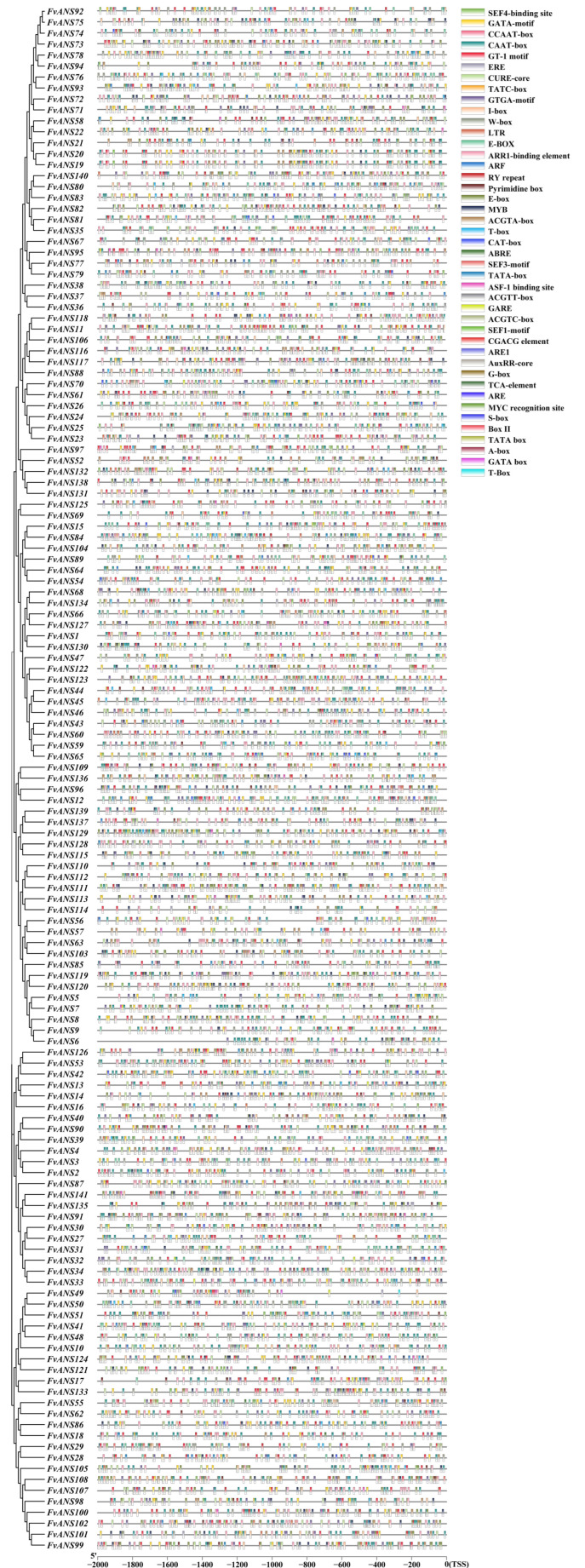
Cis-regulatory element analysis of the *FvANS* genes. Different colors on the right represent different elements.

**Figure 5 ijms-24-12554-f005:**
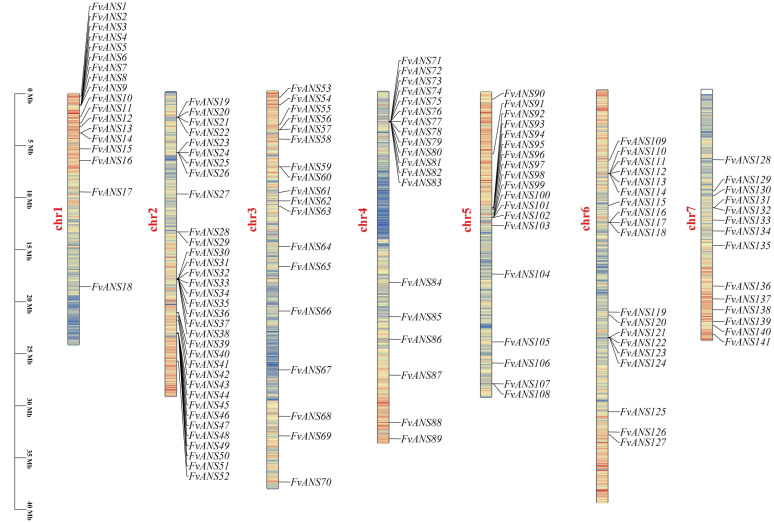
Chromosome distribution of the *ANS* gene family in strawberries. The left scale indicates the chromosome length (Mb), with *ANS* gene markers on the right side of each chromosome. Different chromosomal colors indicate different gene densities, with red indicating the highest density and blue the lowest density.

**Figure 6 ijms-24-12554-f006:**
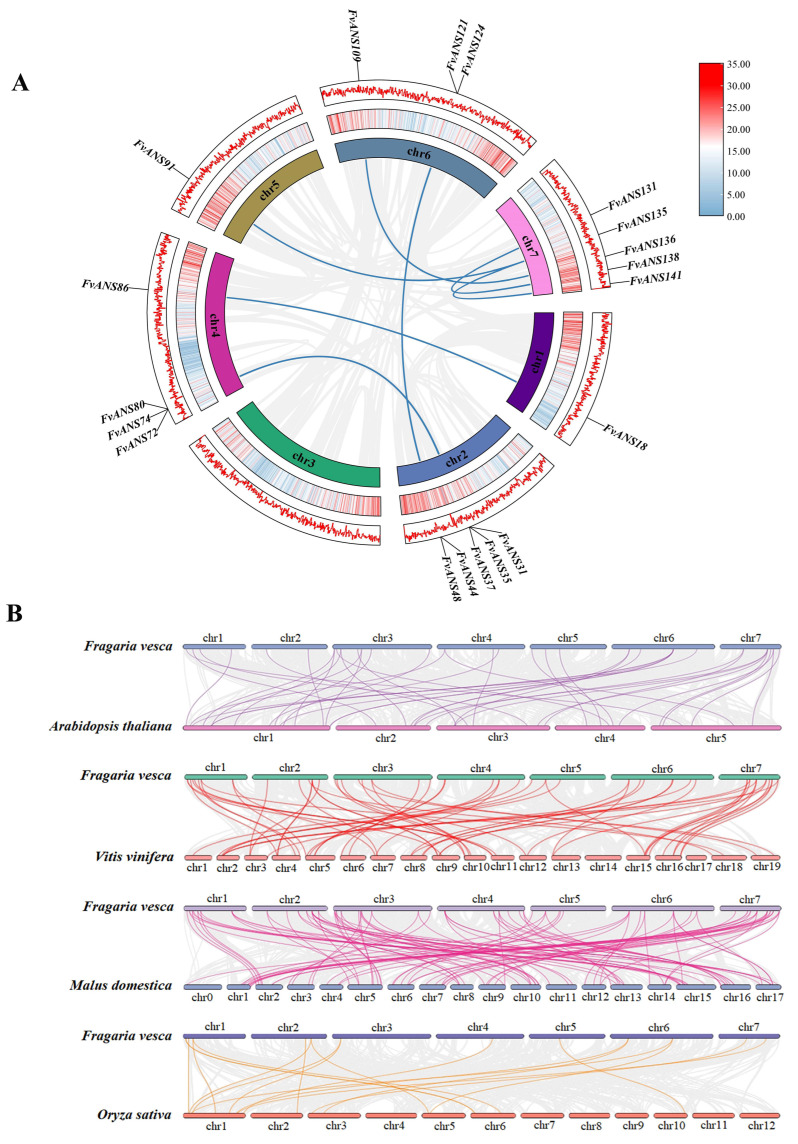
Collinearity analysis of *ANS* gene families. (**A**) Collinearity analysis of *FvANS*. The gray lines represent all collinear blocks in the strawberry genome, and the blue lines represent gene pairs between the *FvANS* genes. (**B**) Collinearity analysis of *ANS* gene in strawberries and four representative plants. The gray lines in the background show collinearity between the strawberry and *Arabidopsis thaliana*, grape, apple and rice genomes. The purple lines show collinearity between the *FvANS* gene and *Arabidopsis thaliana*, the red lines show collinearity between the *FvANS* gene and peach, and the peach lines show collinearity between the *FvANS* gene and apple. The yellow lines represent collinear gene pairs between the *FvANS* gene and rice.

**Figure 7 ijms-24-12554-f007:**
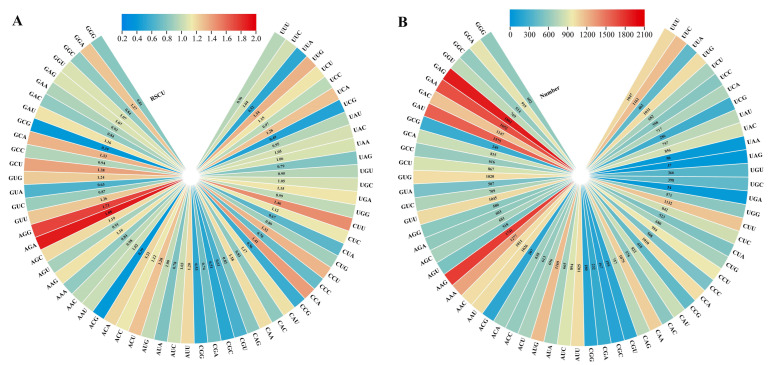
Relative synonymous codon usage and quantity analysis of *ANS* gene codon in strawberries. (**A**) Usage of synonymous codons. (**B**) The preferred number of synonymous codons.

**Figure 8 ijms-24-12554-f008:**
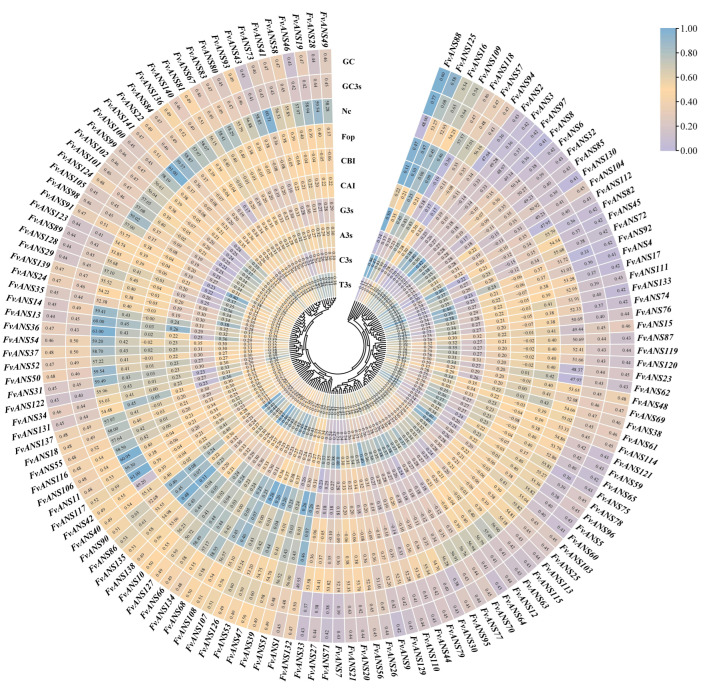
Codon parameter analysis of the *ANS* genes in strawberries. “A3s, G3s, C3s and T3s” refer to the synonymous codon corresponding base frequency on the third; “CAI” refers to the codon adaptation index; “CBI” refers to the codon bias index; “FOP” refers to the frequency of optimal codons; “ENc” refers to the effective number of codon; “GC3s” refers to the amount of the third codon (G + C); “GC” refers to the count of genes (G + C).

**Figure 9 ijms-24-12554-f009:**
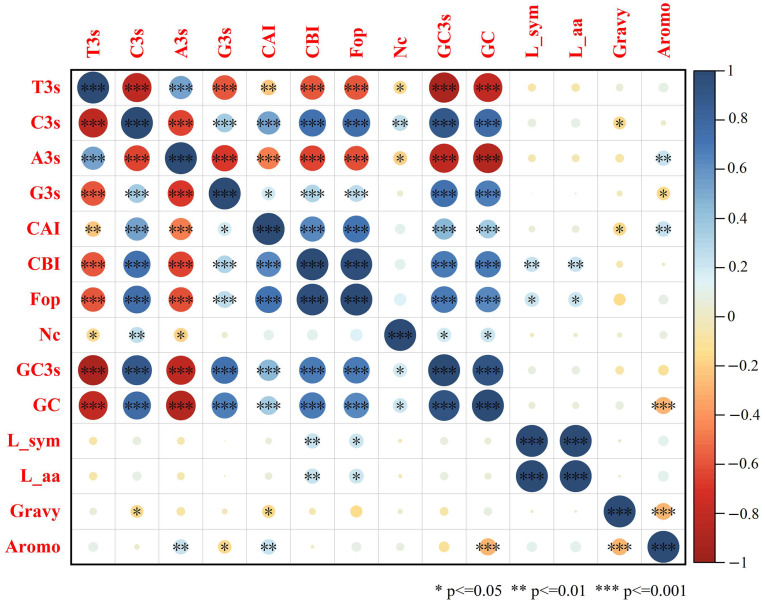
Correlation analysis of *ANS* gene codon in strawberries. Blue indicates positive correlation, red indicates negative correlation and white indicates no correlation. The darker the color, the larger the circle and the stronger the correlation, and vice versa. The number of observations (n) of the correlation coefficient is 141.

**Figure 10 ijms-24-12554-f010:**
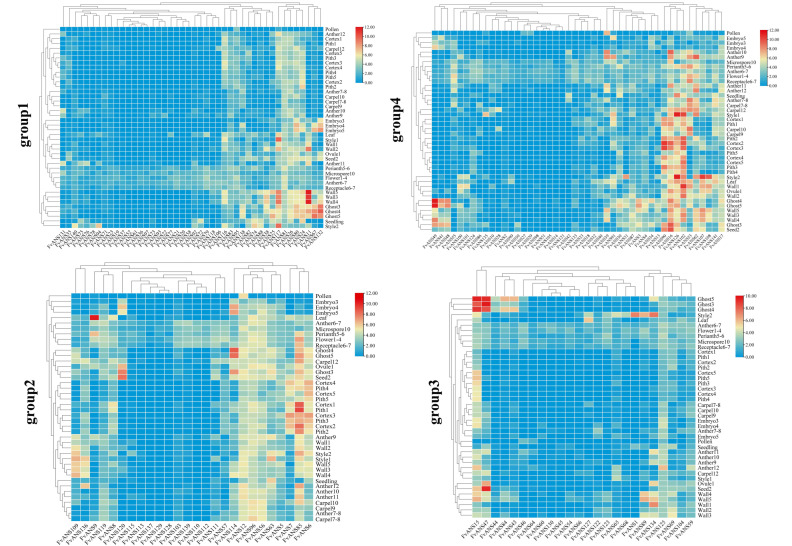
Expression of *ANS* gene in different tissues of strawberries. The numbers behind different tissues indicate developmental stages. Red or blue shading represented the up-regulated or down-regulated expression level, respectively. The scale denoted the relative expression level.

**Figure 11 ijms-24-12554-f011:**
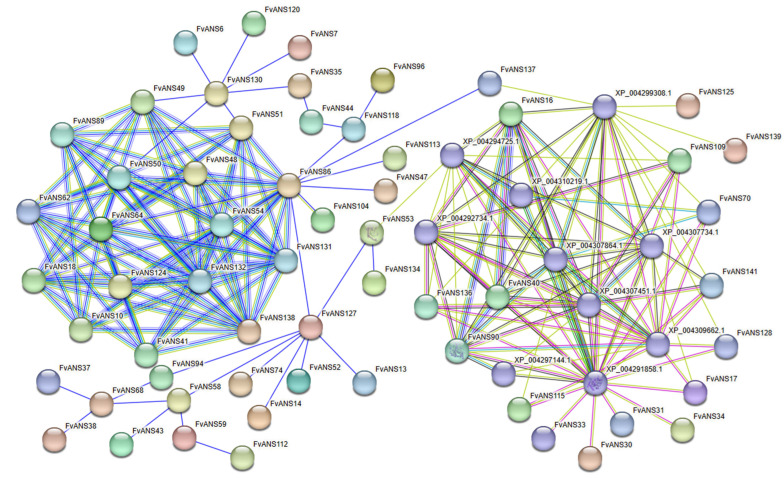
Analysis of protein interaction of *ANS* gene family in strawberries. Nodes indicate proteins. Empty nodes indicate the protein of unknown 3D structures, and filled nodes indicate that some 3D structures are known or predicted. The connection between nodes indicates the interaction between proteins, and different colors correspond to different types of interaction.

**Figure 12 ijms-24-12554-f012:**
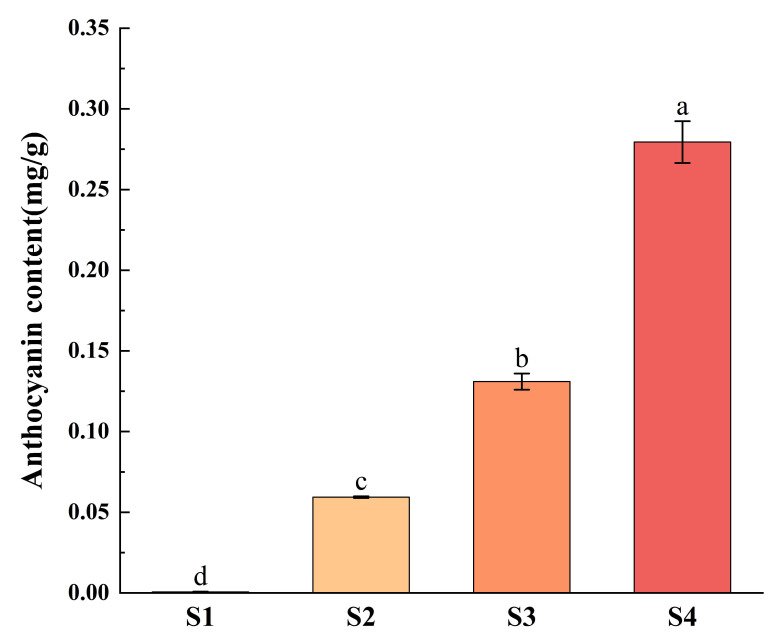
Content of strawberry anthocyanins in four periods. S1 represents the green fruit stage, S2 represents the 20% coloration stage, S3 represents the 50% coloration stage and S4 represents the complete coloration stage. The critical value is 4.07 by checking the F critical value table. Different letters denote significant differences.

**Figure 13 ijms-24-12554-f013:**
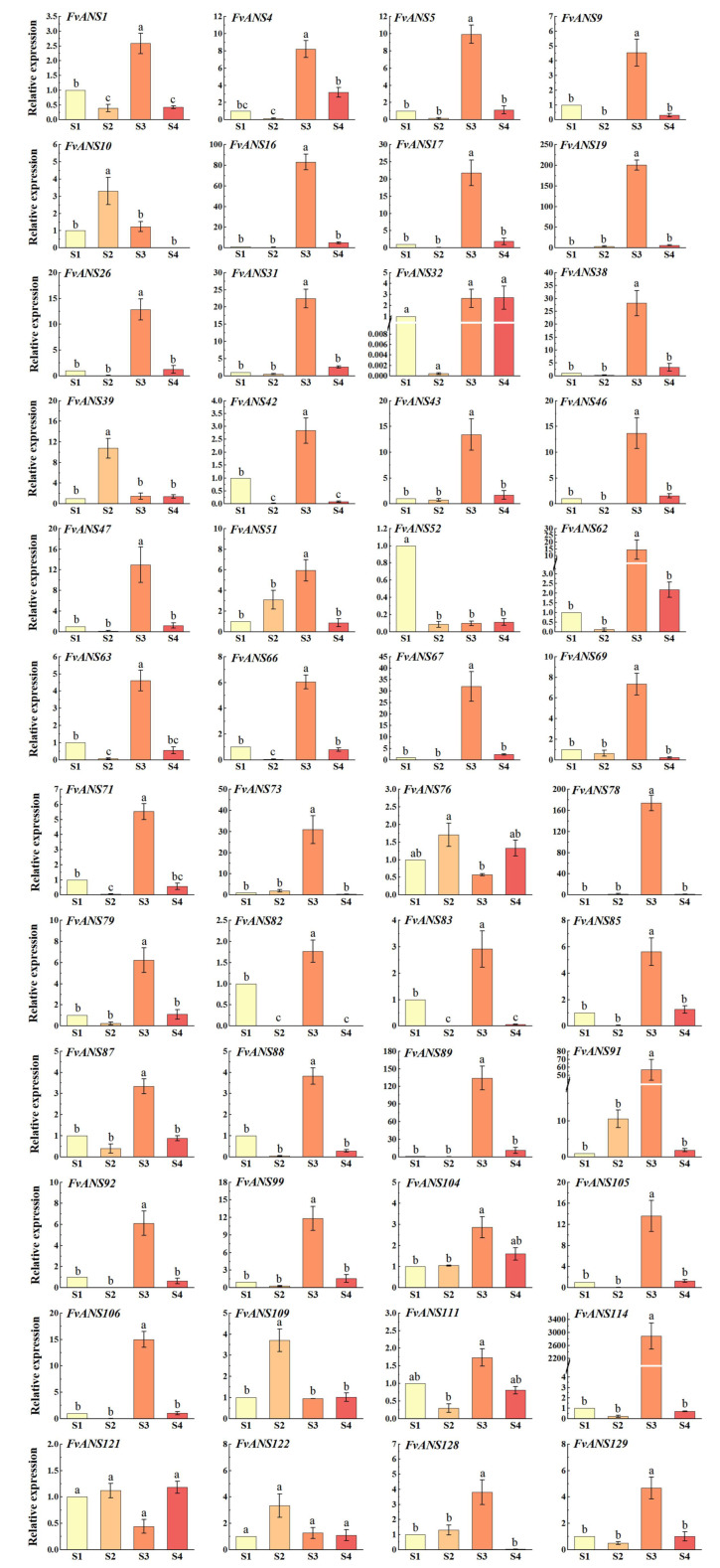
Relative expression levels of *ANS* gene in strawberries treated at different periods. S1 period was used as control. The 2^−∆∆Ct^ method was used to calculate the relative expression. Error bars represent the mean ± SE from three biological repeats. Different letters denote significant differences, whereas the same lowercase letters indicate no statistical difference (*p* < 0.05). The critical value of each gene was 4.07 by checking the F critical value table.

**Figure 14 ijms-24-12554-f014:**
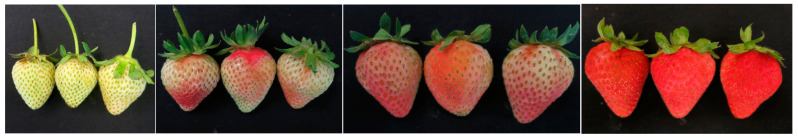
The strawberry has four different coloring stages. From left to right are green fruit stage, 20% coloration stage, 50% coloration stage and complete coloration stage.

**Table 1 ijms-24-12554-t001:** Selection pressure analysis of *FvANS* gene family.

No.	Paralogous Pairs	Ka	Ks	Ka_Ks	EffectiveLen	AverageS-Sites	AverageN-Sites
1	*FvANS18/FvANS86*	0.258660701	2.177082112	0.118810723	957	229.8333333	727.1666667
2	*FvANS35/FvANS74*	0.434230041	3.296855542	0.131710363	1074	245.5833333	828.4166667
3	*FvANS37/FvANS72*	0.448522524	2.547630057	0.176054809	1014	229	785
4	*FvANS91/FvANS135*	0.216861978	2.007919382	0.108003329	1029	239.5833333	789.4166667
5	*FvANS109/FvANS136*	0.237949802	3.15385354	0.075447321	1062	249.5	812.5
6	*FvANS131/FvANS138*	0.225387475	2.712942809	0.083078594	1116	246.1666667	869.8333333

## Data Availability

Data will be made available on request.

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
