# Peer review of "Genome-Wide Identification and Expression Analysis of ANS Family in Strawberry Fruits at Different Coloring Stages"

_ijms, 2023, doi:10.3390/ijms241612554_

Round 1
Reviewer 1 Report
The authors did a decent job to characterize the ANS family in strawberry fruits at different coloring stages. They found 141 ANS genes distributed on 7 chromosomes. Those were mainly located in chloroplasts, cytoplasm, and cytoskeleton. QTR-PCR showed that some of those genes were expressed in different fruit coloring process, especially in the 50% coloring stage.
Although this study provides a way to understand the role of ANS gene family in strawberry coloration I was hoping to see here some functional studies from one of those genes. Without that, this study becomes too technical and lacking in tangible or impactful outcomes.
Please see minor comments below:
- Introduction: include the meaning of S1-S4. The authors mentioned this in the abstract, but forgot to explain the strawberry coloring stages here.
- Conclusions: delete "will" from line 378.
- References: add title to reference 12.
The quality of English language was fine.
Reviewer 2 Report
This article presented Genome-Wide Identification and Expression Analysis of ANS Family in Strawberry Fruits at Different Coloring Stages. The results provide information that different members of the FvANS gene family play a role in different pigmentation stages, and this study lays a foundation for further study of the function of the ANS gene family. Before recommending this article for publication, there are some shortcomings for that should be resolve.
The abstract should provide quantitative results.
Line 8,9 of the abstract “we searched the” where searched?
“and studies have found that they have anticancer function” which studies the sentence should be cited with recent study. The following articles could be cited https://doi.org/10.1016/j.heliyon.2023.e15909, https://doi.org/10.1016/j.foodcont.2022.109496
Second paragraph of the introduction must add how genetic studies such as genome wide identification can be helpful in identifying color specific genes and improving strawberry production.
In many places the authors wrote studies have shown but these sentences lack references. The authors should cite references. Such as line 46-47 should be cited with recent studies https://doi.org/10.1016/j.foodres.2023.112637, https://doi.org/10.1007/s10725-021-00785-7, https://doi.org/10.1002/aoc.5190
Line 73-75 lack references.
Figure 2 resolution is not good words are not readable.
Table 4, instead of this if figure is added with be good.
Discussion section should explain and compare the results.
Conclusion should provide specific insights for the future studies. Like what type of studies are possible based on this study.
Typos and grammatical mistakes should be revise in the whole MS. Avoid long sentences to convey clear message to readers
Reviewer 3 Report
The manuscript entitled “Genome-Wide Identification and Expression Analysis of ANS Family in Strawberry Fruits at Different Coloring Stages” by Feng et al. aimed to elucidate the structural characteristics, phylogeny and biological function of anthocyanin synthase (ANS) genes and their roles in anthocyanin synthesis. The authors have suggested that different members of the FvANS gene family play a role in different pigmentation stages, and this study lays a foundation for further study of the function of the ANS gene family. The experimental design is fine and the authors have experimental support to verify their findings.
However, I have the following major comments for the authors:
1. Table 1, 2 and 3 are quite long. Please place them as supplementary tables.
2. Under the section: 2.3. Gene structure, motif, domain, and promoter cis-acting elements, line 15-17, the authors have mentioned, “The cis-acting element analysis in the first 2000 bp of strawberry ANS genes revealed that FvANS genes mainly contain light, hormone, abiotic stress and meristem response elements.” Please clearly mention regarding the extraction of first 2000 bp promoter sequences. From which site (point) did you extract the genomic DNA sequences? There is no 5’, 3’position, and the nucleotide numbering in the Figure 3 for the promoter cis-element analysis.
In the Material and Methods under the section: 4.5. Analysis of gene structure, motif, domain, and cis-acting elements, line 312-314, the authors have mentioned, The 2000bp upstream sequence of the FvANS gene was obtained using the TBtools software with the online software PlantCARE (http://bioin 313 formatics.psb.ugent.be/webtools/plantcare/html/) and plotted at TBtools (Version 1.108).”
The authors have used Plant CARE database for promoter cis-element analysis. However, this database is very old and does not have up to date information. Hence, you are certainly going to miss many important and new cis-elements that are recently identified. The best option is to use the MATCH program in TRANSFAC (geneXplain) which you need to pay for the subscription. However, the authors can also use PlantPAN and PLACE database if you do not have access to TRANSFAC. PlantPAN is quite up to date.
3. In the discussion, the promoter cis-element analysis part is very brief: only 2-3 lines. This analysis can provide very important information. Please elaborate little longer about the cis-acting elements and their role.
4. Discussion needs major revision. The authors have reviewed information from the literature on ANS gene from different plants rather than discussing their own results. Please critically discuss your results. Is there any relevant information on protein interaction? If so, please discuss about it.
5. Please check the formatting of the references in the main text and also check the language throughout the manuscript.
6. If you have photos of strawberries on different stages of colouration, please provide that.
Please check the language throughout the manuscript.
Reviewer 4 Report
Manuscript "Genome-wide identification and expression analysis of ANS family in strawberry fruits at different coloring stages" is very interesting.
General comments:
Authors identified members of the strawberry ANS gene family from the whole genome of strawberry, and conducted bioinformatics analysis. Real-time quantitative PCR(qRT-PCR) was used to analyze the changes in the expression levels of each member at different periods, so as to clarify the role of each member in regulating the synthesis of fruit anthocyanin, and provide candidate genes for further study of their functions in the future.
Detailed comments:
Figure 1: What method was used to calculate the similarities from which the dendrogram was then constructed?
Figure 2: What method was used to calculate the similarities?
Figure 2: What method was used to construct the dendrogram?
Figure 9: What method was used to calculate the similarities?
Figure 9: What method was used to construct the dendrogram?
The section "4.11. Statistical analysis of the data" is very strangely written. The authors wrote "significant difference." However, they did not indicate the differences between what they meant. Nor did they indicate what method was used to test the significance of the differences. They only stated the level of significance. In addition, there is no information about checking the empirical distribution against the normal distribution. The above analyses should be supplemented.
Section 4.11. is very poor. It should be supplemented with all the statistical methods used in the manuscript. Missing, for example, is information on the method used to test the significance of differences. Also missing is information about the methods I wrote about above. In addition, there is no information about checking the empirical distribution against the normal distribution.
My suggestion:
Figure 8: Add the number (n) of observations from which the correlation coefficients were calculated.
Figure 11: Add the value of LSD or HSD.
Figure 12: Add the values of LSD or HSD.
Paper needs major revision.
Round 2
Reviewer 3 Report
The authors have tried to do some revisions. However, it is not satisfactory.
I still have the following comments for the authors:
1. The authors need to define the promoter sequence. They have again mentioned that “The first 2000bp sequence of strawberry genome was extracted from TBtools for cis-acting element analysis. “ What do you mean by “The first 2000bp sequence’? From which site (point) did you extract the promoter sequences from the genomic DNA sequence? I do not mean the website, it is the promoter sequences: upstream/downstream of which site? You need to specify that. If you have not used the correct promoter sequences, then all your cis-element analysis are not correct.
2. Please revise the nucleotide numbering in the Figure 3 for the promoter cis-element analysis .
3. The authors have used Plant CARE database for promoter cis-element analysis. However, this database is very old and does not have up to date information. Hence, you are certainly going to miss many important and new cis-elements that are recently identified. The best option is to use the MATCH program in TRANSFAC (geneXplain) which you need to pay for the subscription. However, the authors can also use PlantPAN and PLACE database if you do not have access to TRANSFAC. PlantPAN is quite up to date.
4. Discussion still needs major revision. The authors have reviewed information from the literature on ANS gene from different plants rather than discussing their own results. Please critically discuss your results.
5. Please provide the photos of strawberries on different stages of colouration in the main manuscript.
Minor editing required for the manuscript.
Reviewer 4 Report
From the answers provided by the Authors, it appears that they are using the software automatically with no knowledge of the statistical methods used. The authors should contact a statistician.
Point 1: Figure 1: The dendrogram was constructed using the Neighbor-Joining method. However, I asked what method was used to calculate the similarities?
Item 2: Figure 2: The dendrogram was constructed using the Neighbor-Joining method. However, I asked what method was used to calculate the similarities?
Item 4: Figure 9: The dendrogram was constructed using the Neighbor-Joining method. However, I asked what method was used to calculate similarities?
Item 6: The answer that the statistical data was analyzed using Excel software is insufficient. Providing the software is not equivalent to describing the research methods. A scientific publication must be written in such a way that anyone can repeat the experiment under similar conditions. This also applies to statistical analysis, a description of which is still lacking in the manuscript.
Item 8: Figure 11: For each sub-figure, provide a critical value for determining homogeneous groups.
Item 9: Figure 12: For each sub-figure, provide the critical value for determining homogeneous groups.
Round 3
Reviewer 3 Report
Thank you authors for revising the manuscript.
However, I have the following major comments for the authors regarding promoter cis-element analysis:
1. Actually, the promoter sequence is always in the upstream of the Transcription Start Site (TSS). However, the authors have taken 2000 bp upstream of ATG site, which is Translation Initiation Site (TIS). Hence, the sequences they have taken could be a part of Exon 1 and a part of promoter or only Exon 1 depending on the gene length. Hence, the cis-element detected in those regions are not correct, as the sequence is not right. Sometime, some distal cis-elements are present in the downstream of promoter sequence, but majority are present in the upstream part of promoter sequence. Hence, please extract the promoter sequence from the upstream of TSS and redo the cis-element enrichment analysis. Accordingly, you need to revise the methods, results and discussion part that will be based on new promoter sequence analysis.
2. Please revise the nucleotide numbering and 3’ position in the Figure 4 for the promoter cis-element analysis.
Minor editing is required.
Reviewer 4 Report
Now, all is Ok.
Author Response
I would like to thank the reviewers for their suggestions, which have guided my paper writing and scientific work.
Round 4
Reviewer 3 Report
Thank you authors for taking the correct promoter sequences for your analysis.
However there is still some major mistakes in Figure 4. The position of 3' nt and numbering of promoter nucleotides are not correct. Please correct it.
Minor corrections required.
Round 5
Reviewer 3 Report
Thank you authors for correcting Figure 4.
However, there is a minor mistake in the figure. Point '0' is TSS and the downstream nts of TSS are in 3'. Hence, please remove 3' from the figure.
Minor editing is required.
